# Determination of Body Parts in Holstein Friesian Cows Comparing Neural Networks and k Nearest Neighbour Classification

**DOI:** 10.3390/ani11010050

**Published:** 2020-12-29

**Authors:** Jennifer Salau, Jan Henning Haas, Wolfgang Junge, Georg Thaller

**Affiliations:** Institute of Animal Breeding and Husbandry, Christian-Albrechts-University Kiel, Olshausenstraße 40, 24098 Kiel, Germany; jhaas@uv.uni-kiel.de (J.H.H.); wjunge@tierzucht.uni-kiel.de (W.J.); gthaller@tierzucht.uni-kiel.de (G.T.)

**Keywords:** machine learning, neural networks, k nearest neighbour classification, 3D data, dairy cattle, automated body part determination

## Abstract

**Simple Summary:**

The recognition of objects in three dimensional (3D) data is challenging, especially when it comes to the partition of objects into predefined segments. In this study, two machine learning approaches were applied to recognize the body parts head, rump, back, legs and udder of dairy cows in 3D data recorded with Microsoft Kinect V1, for usage in automated exterior evaluation. Five properties of data points were used as features in order to train a k nearest neighbour classifier and a neural network to determine the body part on which the point was located. Both algorithms used for this feature-based approach are computationally faster and more flexible compared to a model-based object detection priorly used for the same purpose. Both methods could be considered successful for the determination of body parts via pixel properties. Reaching high to very high overall accuracies and very small Hamming losses, the k nearest neighbour classification is superior to the neural network which reached medium to high evaluation metrics. However, k nearest neighbour classification is at runtime prone to higher costs regarding computational time and memory, while once trained, the neural network delivers the classification results very quickly. This needs also to be taken into account in the decision of which method should be implemented.

**Abstract:**

Machine learning methods have become increasingly important in animal science, and the success of an automated application using machine learning often depends on the right choice of method for the respective problem and data set. The recognition of objects in 3D data is still a widely studied topic and especially challenging when it comes to the partition of objects into predefined segments. In this study, two machine learning approaches were utilized for the recognition of body parts of dairy cows from 3D point clouds, i.e., sets of data points in space. The low cost off-the-shelf depth sensor Microsoft Kinect V1 has been used in various studies related to dairy cows. The 3D data were gathered from a multi-Kinect recording unit which was designed to record Holstein Friesian cows from both sides in free walking from three different camera positions. For the determination of the body parts head, rump, back, legs and udder, five properties of the pixels in the depth maps (row index, column index, depth value, variance, mean curvature) were used as features in the training data set. For each camera positions, a k nearest neighbour classifier and a neural network were trained and compared afterwards. Both methods showed small Hamming losses (between 0.007 and 0.027 for k nearest neighbour (kNN) classification and between 0.045 and 0.079 for neural networks) and could be considered successful regarding the classification of pixel to body parts. However, the kNN classifier was superior, reaching overall accuracies 0.888 to 0.976 varying with the camera position. Precision and recall values associated with individual body parts ranged from 0.84 to 1 and from 0.83 to 1, respectively. Once trained, kNN classification is at runtime prone to higher costs in terms of computational time and memory compared to the neural networks. The cost vs. accuracy ratio for each methodology needs to be taken into account in the decision of which method should be implemented in the application.

## 1. Introduction

Precision livestock farming (PLF) deals with the task of providing a sustainable food production in sight of a rapidly growing world population ([1]). The goal of PLF is to inform farmers automatically regarding health ([2]) or welfare status ([3]) of their animals based on analysed sensor data.

The development of camera-based monitoring devices has taken a firm place within PLF, as applications of cameras are various and it is a non-invasive method of collecting data from animals. Common problems associated with the usage of camera technology in animal-related settings and barn environments are diffused light conditions, difficult to control animal movement, surface structures that diminish the measurement precision of the camera ([4,5]). However, especially in dairy science, a large number of camera-based studies have been successfully published regarding multiple topics. In lameness detection in cattle 2D cameras ([6,7]), 3D cameras ([8,9,10]) and combinations between cameras and other sensor data ([11]) came to use. Halachmi et al. used thermal cameras in the determination of body condition ([12]), and 2D ([13]) but especially 3D cameras were widely applied to measure the energy reserves of cattle ([14,15,16,17,18]). Cameras have also been applied in animal identification ([19]), the monitoring of herd activities ([20]), animal behaviour ([21,22]) and the animals’ use of barn space ([23]). Salau et al. presented a scanning passage for dairy cows in which six Kinect V1 cameras were mounted ([24]). The goal was an automated conformation recording ([25]), and the linear descriptive traits “udder depth” and “rear leg angle” ([26,27]) were calculated from the 3D data ([28]). The building of the prototype and the acquisition of data happened during the transition from Kinect V1 to Kinect V2. As it was not possible to run several items of the Kinect V2 by one computer, development was proceeded using the Kinect V1 ([29,30]), which works with the “Structured Light” depth measurement principle ([31]) and is in contrast to time-of-flight cameras ([32]) less prone to motion artefacts.

With a lot of sensors, the collected data were measured from the animal directly as the desired target variable, for example activity loggers ([33]). However, looking at camera data, the animal-related information needs at first to be calculated from the raw data (images or videos) instead, which could be challenging. It is a general advantage of 3D cameras that depth value based segmentation can be applied to specify the objects of interest in the foreground. However, the 3D object recognition of individual body parts ([34]) of the recorded cow is a difficult task, while crucial for precise measurements. Salau et al. used an object detection pipeline based on 3D models of the objects of interest (udder, hind leg) to determine the body parts ([28]). Although it served successfully as a basis for the calculation of linear traits, this model-based object detection was very costly in terms of computational time and memory. To process one point cloud could take up to several minutes which was beyond limits for every practical application. Furthermore, an individual model would have been necessary for every additional body part. This posed as a meaningful drawback of the application. The authors were, therefore, looking to replace the model-based approach in favour of an approach based on the features of pixels in the projections of the 3D information into 2D matrices (depth maps). Five features were generated for every pixel on the cow and labelled with the body part on which it was located in order to apply supervised machine learning afterwards.

Machine learning techniques have increasingly been used in animal science. McCulloch and Pitts introduced artificial neural networks as an attempt to model the neural activity of the human nervous system ([35]). Through the repeated processing of numerical information, neural networks ([36]) could learn pattern in the presented data and transfer them to unseen data. Therefore, neural networks are an efficient and flexible method for data evaluation and automation. A lot of problems in agricultural science deal with non-linear relations, a high number of parameters and dependencies on the time of observation. As these problems are, thus, complex and hard to analyse ([37]), neural networks seem to be a suited tool ([38]). For instance, milk yield has been successfully predicted by the use of neural networks ([39]). Furthermore, [40] trained convolutional neural networks to estimate body condition score in cows, and cow behavioural pattern were detected in [41]. However, not only neural networks but also other machine learning algorithms were applied in dairy science. Ref. [8] implemented a decision tree application with regard to features of the moving leg, and dairy cow’s daily eating time was predicted with random forests by [42]. Again, decision tree algorithms ([43]) but also support vector machine methods ([44]) were used in mastitis detection. Ref. [45] provided forecasts of individual milk production via time series ARIMA models. The k nearest neighbour (kNN) principle was introduced by [46] and is based on the idea that samples of the same class are closed to each other in the feature space. In [47], kNN classification was compared to other machine learning techniques in the distinction between lame and non-lame cows and reached approximately 95% accuracy. In addition, a system for automated recognition of feeding and ruminating of dairy cows based on triaxial acceleration data was proposed in [48]. kNN classification was the superior method of all tested machine learning algorithms, as precision and recall for both feeding and rumination were observed larger than approximately 93%.

The present study serves as a complement to the aforementioned studies [24,28], in which a system to automate the evaluation of dairy cows exterior was introduced. Conformation recording is an important basis of modern dairy cow breeding, and is carried out manually and visually by highly trained classifiers, however, it is prone to subjectivity. It has been the goal of the aforementioned study to provide a more objective approach, and the aim of the present article is, to overcome the earlier mentioned limitations of the model-based object detection. To ensure a complete understanding, details on the former camera installation (Section 2.1), recording software (Section 2.2), data collection and the recorded cows (Section 2.3) were given. kNN classification and sequential neural networks with dense layers ([49]) were chosen as machine learning algorithms and were trained on the labelled data set consisting of five pixel properties as features (Section 2.4). The hyperparameters concerning both algorithms were optimised using a grid search, which is explained together with both machine learning methods (Section 2.5). Lastly, a comparison of the algorithms on an unseen hold-out data set based on various evaluation metrics is presented.

## 2. Materials and Methods

### 2.1. Recording Unit with Six Kinect Cameras

Cows were recorded while walking through a framework (Figure 1A) with passage height and width approximately 2 m. On both sides, three depth cameras of type Microsoft Kinect V1 ([29,30]) were installed, such that both sides of the cow were simultaneously recorded. Two Kinect cameras were installed at a height of 0.6 m above the ground with their lines-of-sight parallel to the ground and directed inwards (side view cameras). Pairs of Kinect cameras were placed in the diagonal parts on each side of the framework (top view cameras). As the 57° horizontal field of view (FOV) of a single Kinect camera was too small to capture the cows during several steps, the FOVs of two cameras were combined here. This was achieved in fixing the cameras top to top crossed to each other (Figure 1B), with their bases pointing in opposite directions. This led to one camera mounted upright and one camera placed upside down. In total, for this recording unit three types of camera positions (‘U’, ‘N’, ‘S’) were defined:U: Top view camera; the upper camera in a pair; oriented upside down (base up)N: Top view camera; the bottom camera in a pair; oriented normally (base down)S: Side view camera

For each position, two cameras were present. A 112° combined horizontal FOV could be reached, when the angle between the camera fronts was fixed at ≈56° (Figure 1C).

The Microsoft Kinect V1 camera (PrimeSense, Tel Aviv, Israel) held both an RGB-camera and a depth sensor working with the “Structured Light” principle. The Kinect camera emitted an infra red pattern and detected the deformations in this pattern to measure the distances between sensor and objects. These depth data were provided as depth maps of 640 × 480 pixels resolution.

At the research farm Karkendamm of Kiel University (Northern Germany), this recording unit was firmly installed in a separate room next to the loose-housing barn. As in detail described and depicted in [50], the room held a solid round tour so that cows did not need to be led through the framework by halter, but freely walking cows could be recorded for repeated runs.

For Kinect cameras to be recognised as individual devices by a single computer, each camera needs to be connected to its own USB controller. A computer with an AMD FX-8350 CPU clocked at 4.0 GHz on an ASUS M5A99FX PRO R2.0 mainboard and Kingston Value RAM ECC-DDR3 clocked at 1.6 GHz (PC3-12800) was thus equipped with USB cards to accommodate six Kinect cameras.

### 2.2. Recording Software

Based on OpenNI ([51]), software to operate multiple Kinect devices at the same time, a data format to store the depth maps as binary video streams and software to view the video streams were developed. Furthermore, synchronisation of the cameras ([24]) and extrinsic calibration of the recording system was taken care of. As an initialisation for the usage of a multi-camera system, calibration is essential. Intrinsic parameters are the focal length of the camera, scaling, radial bias, and the offset between image centre and origin of the coordinate system in the image plane ([52]). These have been specified ([53]) and used in the calculation of 3D coordinates of the points in the scene according to [54]. Extrinsic parameters are given as rotations and translations between coordinate systems of different cameras or between the camera and a global reference frame. To achieve this, a calibration object was introduced in [55], and recorded by all six cameras individually prior to recording cows. In [55], it was also described in detail how the necessary rotations and translations were calculated from the recordings of the calibration object. Furhtermore, the system recorded empty scenery for a specified lead time, and the averaged background image was used for segmentation. Those areas that show large pixel-wise differences to the averaged background could be specified as moving foreground ([56]). Using wavelet decompositions, it was shown in [57] that the background regions of the images contained more high frequency parts than the foreground regions. This was used to refine the segmentation results. Background pixel was set to zero. More detailed information on the specification of threshold, the estimation of a background image and the selection of cow images could be found in [28].

### 2.3. Data Collection and Recorded Cows

Between November 2015 and April 2016, 18 Holstein Friesian cows were recorded. The cows were in lactations 1 to 5, with the milk yields ranging from 10.6 kg to 32.7 kg (22.2 kg ± 6.8 kg). Cows’ weights after morning milking on the days of recording varied from 540 kg to 835 kg (680.9 kg ± 78.3 kg), and the sacrum heights ranged between 1.43 m and 1.49 m (1.46 m ± 0.02 m). Cows passed the framework repeatedly and cameras were not turned off between consecutive runs, but all six cameras were simultaneously acquiring depth maps. Thus, recording empty scenery could not be avoided.

The disturbance of the animals was kept to a minimum and recording by camera is a non-invasive method of collecting data. The authors declare that the “German Animal Welfare Act” (German designation: TierSchG) and the “German Order for the Protection of Animals used for Experimental Purposes and other Scientific Purposes” (German designation: TierSchVersV) were applied. No pain, suffering or injury was inflicted on the animals during the experiment.

### 2.4. Preparation of a Data Set of Pixel Properties

The helper tool ‘Polygon drawer’ was self-developed in MATLAB and used to generate a training data set. Overall, 100 depth maps for each camera were randomly selected, i.e., 200 depth maps for each camera position (‘U’, ‘N’, ‘S’). One person manually labelled the body parts head, rump, back, the foreleg facing the camera, the foreleg avert from the camera, the hindleg facing the camera, the hindleg avert from the camera and the udder by mouse-clicking polygons around them (Figure 2). Hereby, the body part ‘udder’ was excluded with regard to camera position ’U’, as udders were not seen from this camera position. Thus, in the aspired multiclass classification, these eight, respectively, seven body parts were used as responses.

From all inside pixels of the polygons, five properties were listed as features (predictors) to be used in machine learning. The first three were row index (‘row’), column index (‘col’) and depth value in mm (‘depth’). Additionally, the mean curvature ‘m_curv’ was used. The mean curvature is the average of the principal curvatures ([58], Figure 3) and can be calculated from the surface gradient and the Hessian matrix in the surface point. A nine-pixel-square surrounding the respective pixel was used to determine the gradient g and Hessian matrix H, which were inserted in the following formula:(1)$$m\_\mathrm{curv}\;=\;\frac{gHg^{T}-{\left|g\right|}^{2}\;\mathrm{Trace}\left(H\right)}{{2|g|}^{3}},$$
with Trace(H) being the sum of the diagonal elements of H. As a last property, the variance of depth values (‘var’) was calculated from the nine-pixel-square surrounding the respective pixel.

The body part to which the pixel belonged was one-hot encoded (Table 1). One list per camera position was generated, resulting in 915,587 rows for camera position ‘U’, 1,295,302 rows for camera position ‘N’, and 1,352,017 rows for camera position ‘S’. The data were afterwards standardised with the MinMaxScaler from the preprocessing sub-module of Python module scikit-learn ([59,60]), to ensure comparable scales of the features. A 30% hold-out data set was split off from all three data sets.

#### 2.4.1. Balancing the Data Set Using Synthetic Minority Oversampling Technique (SMOTE)

As the udder or the legs naturally take fewer spaces in the images than the rump or back of a cow, the data set was hardly unbalanced, thus, an oversampling technique was applied. The synthetic minority oversampling technique (SMOTE) was introduced by [61]. Additional samples $x_{new}$ are generated to balance the classes by interpolating between an existing sample $x_{i}$ and one of its nearest neighbours $x_{j}$ via $x_{new}=x_{i}+\lambda *\left(x_{j}-x_{i}\right)$, λ∈[0,1] being a random number. SMOTE was applied to the respective training data (Section 2.5.1 and Section 2.5.2) using the Python module ‘imbalanced-learn’ ([62]).

### 2.5. Applied Machine Learning Methods

The two machine learning methods kNN classification and neural networks were used in multiclass classifications with eight (seven with camera position ‘U’) responses and five features (Section 2.4). The methods were applied to all camera positions separately. In machine learning, hyperparameters are the model parameters which are not learned during training, but need to be set to control the learning process. For both methods and all camera positions, separate hyperparameter tunings were performed using a grid search. Ranges for all hyperparameters had to be predefined by the user. The parameters were then optimized over the resulting parameter grid.

The hyperparameter tuning as well as training and application of the models were run on an Intel^®^ Core^TM^ i7-7500U CPU clocked at 2.7 GHz using a NVIDIA GeForce 940MX GPU.

#### 2.5.1. k Nearest Neighbours Classification

kNN classification ([46]) is a non-parametric method of supervised learning with the hyperparameter k being the number of neighbours used for classification. With kNN classification, the labelled data points are stored during training, and every unlabelled data point is then classified according to the most frequent class among its k nearest neighbours. For continuous variables, the nearest neighbours are commonly determined from the Euclidean distance.

A grid search was performed for values of k ∈{1,…,20} using GridSearchCV from the model selection sub-module of scikit-learn with five-fold cross-validation on the training data set. The training data set was randomly split into five equally large parts. Successively, the kNN classifier was trained on four parts and evaluated on the left out fifth part. Prior to training, SMOTE (Section 2.4.1) was applied to balance the training data. As evaluation metric, the accuracy on the left out part was calculated. The average of these accuracy values was used to evaluate the choice of the value for the hyperparameter k. When averaged accuracy was equal for different values of k, the smallest k was chosen.

#### 2.5.2. Neural Networks

Artificial neural networks ([36]) originated from the attempt to model the neural activity of the human nerve system ([35]). Similar to the human brain, artificial neural networks are able to learn an underlying pattern through the repeated processing of numeric data.

The Keras application programming interface ([49]) was used with a Tensorflow backend (Tensorflow 1.14.0; [63]) to model and train neural networks. The neural networks comprised a plain stack of dense layers, i.e., as input for each layer the output of the former layer is used. This has to be considered in contrast to more complex nerual network models, in which layers share input or multiple output is generated by some layers. The input layer of all models held five neurons, one for each pixel property (Section 2.4, Table 1). The output layer of all models held seven (camera position ‘U’), respectively, eight nodes (camera positions ‘N’ and ‘S’) according to the number of body parts to distinguish between. The hyperparameters ‘numbers of hidden layers’ and ‘number of neurons’ in the hidden layers were optimized with regard to the validation loss within the following ranges:$$\begin{array}{ccc} {}^{\prime }\mathrm{number}\;\;\mathrm{of}\;\;\mathrm{hidden}\;\;{\mathrm{layers}}^{\prime } & \in & \left\{0,\ldots ,16\right\}\\ {}^{\prime }\mathrm{number}\;\;\mathrm{of}\;\;{\mathrm{neurons}}^{\prime } & \in & \{10,50,100,150,200,250,300\}. \end{array}$$

These ranges were chosen to provide a wide variety of fully connected neural networks, as the needed capacity for the problem at hand was unknown. Neural networks were trained and evaluated for all pairs of hyperparameter values. The neural networks with the smallest observed validation loss were chosen for all camera positions. A fifteen percentage dropout ([64]) was implemented as every second layer. Dropout is a method to regularise the learning process and to make the model less prone to overfitting. A predefined percentage of neurons is temporarily removed from the network and does not participate in the current calculation step. Overall, 30% of the training data set was split off for validation. SMOTE (Section 2.4.1) was applied to the remaining 70% prior to training.

The neural networks were trained using the back propagation algorithm. The Adam optimization algorithm ([65]) and the categorical cross entropy loss function were fixed for all neural network models, as the presented problem is a multi-class classification task. In training neural networks, the data are presented to the model several times in small packages (batches) of differening constellations of data points. When the whole training data set was presented to the network in batches, it is called an epoch. Early stopping after five epochs without improvement regarding the loss on the validation data as well as a scheduler to linearly decrease the model’s learning rate with increasing number of trained epochs were implemented for every neural network model.

#### 2.5.3. Evaluation Metrics

The most successful classifier concerning both machine learning methods was evaluated on the hold-out data-set.

Confusion matrices as shown in Table 2 were generated in a ‘one versus rest’ manner for all body part classes using scikit-learn. The ‘one versus rest’ strategy considers the samples of one class as positives and all samples of the other classes as negatives and reduces the problem to evaluate a multiclass classifier to the evaluation of a collection of binary classifiers. True positives *TP*, false positives *FP*, true negatives *TN*, and false negatives *FN* were counted.

##### ‘One Versus Rest’: Precision, Recall, F1-Score

Precision (Equation (Equation 2)) is the percentage of correctly positively labelled samples of all positively labelled samples. Recall (Equation (3)) is the percentage of correctly positively labelled samples compared to all positive samples. The F1-score (Equation (4)) is the harmonic mean of precision and recall and evaluates whether a high value of precision, respectively, recall is due to a low value of the other metric ([66]).
(2)$$\begin{array}{ccc} \mathrm{Precision} & = & \frac{TP}{TP+FP} \end{array}$$
(3)$$\begin{array}{ccc} \mathrm{Recall} & = & \frac{TP}{TP+FN} \end{array}$$
(4)$$\begin{array}{ccc} F1-\mathrm{score} & = & 2*\frac{\mathrm{Precision}*\mathrm{Recall}}{\mathrm{Precision}+\mathrm{Recall}} \end{array}$$

Precision, recall and F1-score were calculated for each body part class separately using the ‘one versus rest’ approach.

##### Overall Metrics: Accuracy and Hamming Loss

The evaluation metrics accuracy and Hamming loss (Equation (Equation 5)) were calculated to measure the performance of the multiclass classifier. Accuracy is the percentage of correct classifications. The Hamming loss is calculated as the averaged Hamming distance ([67]) between the prediction $\hat{y}$ and the actual outcome *y*. With *N* being the number of classes and $1_{X}$ being the indicator function of a set *X*, the Hamming loss $\mathrm{Ha}.\mathrm{loss}(\hat{y},y)$ calculates as the sum
(5)$$\mathrm{Ha}.\mathrm{loss}\left(\hat{y},y\right)=\frac{1}{N}\sum_{j=1}^{N}1_{\left\{\hat{y_{j}}\neq y_{j}\right\}}.$$

##### Kruskal–Wallis Tests

The Kruskal–Wallis test is a non-parametric version of the one-factor analysis of variance. Its prerequisites only include that the dependent variable is measured on an ordinal scale and that the independent variable defines two or more groups free from contradiction. As the homogeneity variance—as an assumption of the analysis of variance—was not given, Kruskal–Wallis tests were conducted for all ‘one versus rest’ evaluation metrics, whether or not the camera position or the machine learning method had an effect.

## 3. Results

Figure 4 presents the accuracy averaged among the five folds of the cross-validation used to evaluate the choice of the hyperparameter k in kNN classification (Section 2.5.1). k = 1 was chosen for camera positions ‘U’ and ‘S’, and k = 3 was chosen for camera position ‘N’. While the averaged accuracy stayed above 0.97 and 0.94 for camera positions ‘U’ and ‘S’, for camera position ‘N’, the observed values dropped from 0.90 to 0.8 quickly with increasing k. The course for camera position ‘N’ shows greater variation from value to value compared to the courses associated with camera positions ‘U’ and ‘S’.

The smallest validation loss with regard to camera position ‘U’ was observed for the neural network, with ten hidden layers and 300 nodes per hidden layer. For camera position ‘N’, a neural network with nine hidden layers and 300 nodes per hidden layer was chosen. Regarding camera position ‘S’, the validation loss reached its minimum for a neural network with four hidden layers and fifty nodes per hidden layer. Figure 5 presents the training curves for all three camera positions.

In Table 3, the overall evaluation metrics accuracy and Hamming loss as well as the ‘one versus rest’ evaluation metrics precision, recall and F1-score for both classification methods can be found. The Kruskal–Wallis test showed that the ‘one versus rest’ metrics were significantly higher (*p* < 0.001) for kNN classification than for classification by neural networks.

The kNN classifiers reached very high accuracies from 0.888 (‘N’) to 0.976 (‘U’) and showed very small Hamming loss values (0.007 to 0.027). The ‘one versus rest’ metrics significantly depended on the camera position (*p* < 0.01), but reached very high values for ‘U’, ‘N’, and ‘S’. Except for the precision associated with rump (0.87), back (0.77), facing foreleg (0.89) and udder (0.84) for camera position ‘N’, all precision values were greater or equal to 0.90 (0.93 ± 0.08). The recall values were greater or equal to 0.93 (0.95 ± 0.05), with the exceptions rump, facing foreleg, facing hindleg and udder for camera position ‘N’, which ranged between 0.83 and 0.88. The F1-scores ranged between 0.84 and 1.00 (0.94 ± 0.06).

The neural networks reached medium-to-high accuracies from 0.684 (‘N’) to 0.841 (‘U’) and showed Hamming losses smaller than 0.08. The camera position had no significant effect (*p* < 0.05) with regard to the ‘one versus rest’ metrics. The precision values were also medium to high, ranging from 0.63 to 0.97 (0.77 ± 0.13), except from the precision values for rump (0.55), facing foreleg (0.58) and udder (0.53) for camera position ‘N’. The range of the recall values was even higher, reaching from 0.45 to 1.00 (0.76 ± 0.17, exceptions not named). The F1-scores ranged between 0.50 and 0.99 (0.76 ± 0.14).

## 4. Discussion

In this article, the machine learning methods kNN classification and neural networks have been applied to the properties of pixel in Kinect depth maps to determine on which body part of a cow the pixel was located. The presented recording unit has in the aforementioned work on automation of conformation recording ([28]) been successfully applied to measure the conformation traits “udder depth” and “rear leg angle”. The recognition of body parts, however, has been dealt with via an object recognition pipeline based on models of the respective body parts, i.e., udder and hindleg. The model-based object recognition has only been applied to the sideview cameras and only to udder and hindleg. Additional models would have had to be used with every additional camera position or body part. Furthermore, the approach was computationally expensive, varying strongly with the processed point cloud. As a calculation, up to several minutes per point cloud was beyond limits for every practical application, the present article introduces a faster and more flexible solution with regard to the body part detection, and a larger variety of body parts are determinable for all three camera positions. For the recording unit, the Kinect V1 was chosen in favour of the newer Kinect V2. This decision was due to technical reasons, as it was not supported to run several devices of Kinect V2 on the same computer. Furthermore, in contrast to the Kinect V1, the Kinect V2 did not feature an acceleration sensor which was crucial for the calibration of the system [55]. As Kinect cameras also detect the pattern projected by another Kinect camera device, interferences needed to be avoided in a multi-Kinect setting. Regarding the side view cameras (camera position ‘S’), the infrared pattern of the opposite camera was occluded by the animal, and interferences on the cow’s surface were prevented when a cow was walking through the passage. The same applied for the infra red pattern of top view cameras from different sides of the recording unit. Interferences between the two top view cameras in a camera pair were reduced to the negligible fraction of 2° overlap between FOVs (Figure 1).

Features in machine learning are quantifiable predictive variables by which the desired outcome is characterised. In this application, five features are used. Row and column index localise the pixel in the depth map. These were considered, because it was likely for some body parts to occur in repeating pixel ranges, as exemplarily of the udder in the recordings of sideview cameras, which were mostly depicted in the upper half of the depth map (Figure 2). The depth value was chosen as a feature, because differing body parts were recorded in different distances to the camera, i.e., back and rump or averted and facing legs. The mean curvature and the variance describe characteristics of the recorded surface, as it was assumed that both the surface curvature as well as the texture differed between body parts. For comparability, the features were standardised prior to training, and to balance the classes, synthetic samples were generated using SMOTE (synthetic minority oversampling technique). As SMOTE creates new, randomly tweaked observations as linear interpolations of existing samples, it was very important to apply SMOTE only to the training data. A split into training and validation data set after the oversampling would have made it possible that newly generated samples were assigned to the validation data, while the original samples were assigned to the training data set. This would have caused the leakage of information from the training into the validation data set and reduce the models capability to generalise to unseen data.

With kNN classification and neural networks, two supervised learning non-parametric algorithms were chosen in order to avoid assumptions for underlying data distribution, but to determine the structure of the machine learning models solely from the data set. This is advantageous regarding the application to real world data sets, which often do not follow mathematical assumptions. High accuracy in classifying the pixel was aspired to cleanly partition the cow into the respective body parts, whereas interpretability was not considered important in this case, as inference was not the goal. This led to the choice of highly flexible algorithms that were not restricted to a specific shape, i.e., a hyperplane as with support vector machines, of the mapping function. Furthermore, the number of features was very low compared to the large number of samples. As variance is the sensitivity of the model towards changes in the features, a large amount of samples in combination with few features makes it likely that models prone to high variance learn the pattern in the data instead of the noise, because they have lower asymptotic error.

Although larger values of k reduce the effects that noise in the data has on the classification ([68]), the boundaries between classes become less distinct. Therefore, in case the cross-validation grid search for the optimal hyperparameter k led to coinciding maximum values for the averaged accuracy (Section 2.5.1), the smallest value for k was used for training the algorithm. Generally, odd values for k are chosen over even values, as with even values, tie situations can occur during the classification process. This might be the reason why in Figure 4 the courses accuracy associated with camera positions ‘N’ and ‘S’ are going up and down for odd and even values for k. For camera positions ‘U’ and ‘S’, even k = 1 was chosen, meaning that every unlabelled data point was classified according to its nearest neighbour. This seemed to be prone to large classification errors, but given a large number of data points—as was the case in this study—chances were high that the unlabelled point and its nearest neighbour joined classes. As discussed in [69], the error associated to this nearest neighbour rule is bounded by twice the Bayes error, with the Bayes error being the lowest possible prediction error that can be achieved if one could know exactly what process generates the data.

Non-linear relations and a high number of parameters make a lot of problems associated with animal science complex and hard to analyse ([37]). In addition, those problems often depend on the observation time. The flexibility of neural networks and their ability to learn various hidden pattern in the data make this method a suitable tool for application in agricultural science ([38]). With neural networks, the number of hidden layers (depth) and the number of nodes (height) are the two main hyperparameters with regard to model capacity. The capacity needs to be chosen high enough so that the neural network would be capable of learning the underlying pattern in the data and would not underfit. However, too much capacity causes the model to memorise the data set including the noise, and to fail with the generalisation on unseen data. The model might also get stuck during optimisation process, when the capacity was set too high.

For the three camera positions (‘U’, ‘N’, ‘S’), neural networks of differing depth and height were chosen, but with ten hidden layers at maximum, none of the architectures could be considered explicitly deep. It was noteworthy that the capacities of the neural networks specified for camera positions ‘N’ and ‘U’ are similar, but more complex than the capacity with regard to camera position ‘S’. A reason for this might be that the relation between cameras and object were similar between camera positions ‘N’ and ‘U’, however, cameras in position ‘S’ had shorter distance to the objects and a horizontal line of sight, while the lines of sight in positions ‘N’ and ‘U’ were diagonal. This might have led to more variability in the recorded 3D data concerning camera positions ‘N’ and ‘U’ and, thus, to more complex neural networks. The capacity of neural network architectures is still an important research topic ([70,71,72]), and its estimation and comparison via quantitative methods are still open problems. Baldi and Vershynin ([70]) state that “everything else being equal”, the functions computed by models with high capacity are “more regular and interesting”. As could be seen in Figure 5, the loss calculated on the validation data mostly was lower than the loss calculated on the training data. This might at first not be intuitive, as in machine learning, the model parameters are adapted to fit the training data set and, thus, the model should have a weaker performance on the validation data that have not been seen during training. Leakage of information from the training data into the validation data set has not taken place, as the data was split prior to the application of SMOTE (Section 2.5.2). Instead, the dropout regularisation could be one reason for the validation loss being smaller than the training loss. A randomly chosen 15 percent of the nodes of a layer were ignored per epoch, and this training accuracy was sacrificed to avoid overfitting. However, during validation, the dropout regularisation was not active, i.e., all nodes contributed to the classification of the validation data set samples. Another reason might be that the validation loss is calculated at the end of the epoch, while the training loss is averaged over the course of an entire epoch, and therefore, measured half an epoch earlier than the validation loss. Looking at Figure 5, the validation loss approaches the training loss or even becomes larger, when shifted half an epoch to the right.

In view of the ’No Free Lunch’ Theorem by [73], both methods would be equivalent given the theoretical situation of a noise-free scenario with misclassification rate as loss function. However, the kNN classification showed very high accuracies and outperformed the neural networks concerning all camera positions and evaluation metrics. Neural networks enable highly complex decision functions, but the kNN classification might have been superior here, because the features describe the position of the pixel in a three dimensional space (row index, column index and depth) with two additional properties (variance and curvature). Therefore, the features could be better modelled as a classification problem that uses Euclidean distance.

Although accuracy is a very common metric and easy to understand, concerning the training of a data set with very unbalanced classes, it can be misleading, as classifications ignoring classes with only few observations could lead to high accuracies. The ‘one versus rest’ metric F1-score was also considered prone to doubtful assessment when it comes to unbalanced classes, as true negatives are ignored ([74]). However, the generalisability of the models trained in this work has been enhanced by balancing the data set, and further evaluation metrics have been calculated to assess model performance in addition to accuracy and F1-score. As the Hamming loss punishes every deviation regardless of the size of classes, and was observed to be very small for kNN classification and neural networks, both methods could be considered successful with regard to the classification of pixel. The ‘one versus rest’ metrics, the precision values, recall values and F1-scores were not only higher but also a lot more homogeneous for kNN classification. As a result of the high variation within camera positions, no significant differences between camera positions could be observed for these evaluation metrics when it came to the neural networks. In contrast, the camera position ‘N’ performed in a significantly inferior manner compared to camera positions ‘U’ and ‘S’ regarding kNN classification. The differing behaviour associated with the data set gathered from camera position ‘N’ was also reflected in the choice of a higher hyperparameter k compared to the other two data sets. Furthermore, overall accuracies for both classification methods were lower and Hamming losses were higher for position ‘N’, and the course of the loss function during the training of the neural network for position ‘N’ showed significantly stronger variation. A reason for this might have been that one of the Kinect devices in camera position ‘N’ was a different model generation of Kinect V1 (model 1414) than the rest of the used devices (model 1473), although all cameras had been ordered at the same time. Information on explicit technical differences between the model generation had not been available to the authors. Although there had been both no differences in handling and no observable changes in image quality, this device might have come with differing depth resolution or variation, resulting in a less homogeneous data set associated with camera position ‘N’.

The system proposed in [28], which was complemented in the present article, could be transferred to different cattle breeds, however, it is likely that the training of the machine learning algorithms would have to be repeated to adapt to the differences caused by the breed-specific exterior. As the background was set to zero during pre-processing (Section 2.2), background noise has not been an issue in the processing of the cow surfaces. However, the depth resolution of the Kinect V1 and variation in depth measurement needed to be considered a limitation of the system when it came to data quality, and the quality of the predictors calculated from the depth data. At the time when the proposed system had been developed, Kinect V1, Kinect V2 and the Xtion Pro Live determined the low cost sector in 3D sensors. For the multi camera system, Kinect V2 had to be excluded for the stated technical reasons, however, it has been used in single camera studies in the estimation of animal dimensions ([75]), where the improved depth resolution facilitated the improvement of state of the art models for withers height, hip height and body mass. The Xtion Pro Live and Kinect V1 had equal depth resolution. Thus, a higher depth resolution had not been in reach when the authors developed the presented system. Meanwhile, the camera technology has undergone further development, and Le Cozler et al. proceeded with the idea of a multi camera framework to evaluate cattle physiology ([76,77]) using higher depth resolution in the commercially available product Morpho3D. Nevertheless, it was not the aim of this study to further develop the system, but to focus on the already collected data and to improve the object detection within the given situation by carefully choosing machine learning models suitable to interpret the given data.

Numerous experts on machine learning have dealt with the question of whether having more data beats fancier algorithms ([78,79]). Thus, the availability of higher resoluted and more 3D data might have enabled both better insights and applications. However, it would be a too simplistic view to look for limitations of PLF systems only in the quality of the data, as this study has proven that still significant differences in performance between algorithms are revealed. Here, the complex model, which is prone to performance improvement when more data is provided, has been outperformed. However, to answer the question of which classification method to use for the determination of body parts, one should not only consider the evaluation metrics given in Table 3. The kNN classification has minimal training costs, as the whole labelled data set is remembered, but classifying a new data point is costly in terms of time and memory, as the decision making is based on the computation of many distances. The neural networks come at very high costs in the training phase, while once trained, only the learned weights need to be applied to classify a new data point. Thus, not only the pure performance, but also applicability and implementability, play an important role in the choice of the machine learning algorithm.

## 5. Conclusions

Dairy cows were recorded while walking through a framework with three Kinect V1 cameras installed on both sides of the framework. The cameras were mounted in three different camera positions. In-depth maps recorded from all camera positions the body parts head, rump, back, forelegs, hindlegs and udder were manually cut out. For the respective pixels the features row index, column index, depth value, as well as variance and mean curvature of the surrounding cow surface were calculated. The data sets for all camera positions were standardised. For kNN classification and neural networks, hyperparameter were specified, and both machine learning algorithms were trained to determine the body part on which a pixel was located based on the aforementioned features. Synthetic minority oversampling technique was applied to the labelled training data set. Compared to a model-based object detection, the feature-based approach is computationally faster and more flexible, as it is not necessary to provide models for all body parts and camera positions. kNN classification reached high to very high overall accuracies and very small Hamming losses. In addition, the ‘one versus rest’ precision values, recall values and F1-scores were high to very high. The neural networks reached medium to high evaluation metrics. Both methods could be considered successful for the determination of body parts via pixel properties, but kNN classification was clearly superior. However, as this method comes with higher costs in terms of computational time and memory compared to the neural networks, not only the performance but also the applicability should be considered in the choice of method. An adaption of the system for different breeds would make a re-training of the machine learning models necessary. This study highlights that not only the amount and quality of data could limit the development of PLF systems, but that instead the performance in the given situation could be significantly improved by choosing and fine tuning the machine learning algorithm with all care.

## Figures and Tables

**Figure 1 animals-11-00050-f001:**
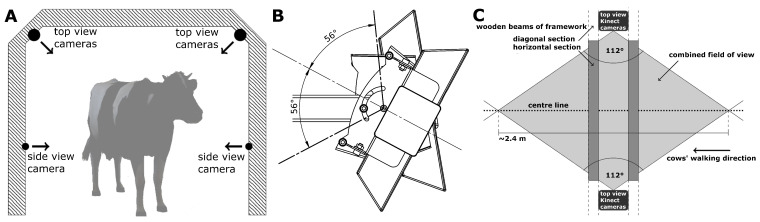
(**A**): Schematic representation of the framework with the positions and lines of sight of the side view and top view Kinect cameras shown as black bullets and arrows. (**B**): Technical drawing of a pair of top view Kinect cameras, fixed with 56° angle between the fronts. In the foreground, the bottom of a Kinect base is seen as pointing away from the pair. (**C**): Combined horizontal field of view of all four top view cameras.

**Figure 2 animals-11-00050-f002:**
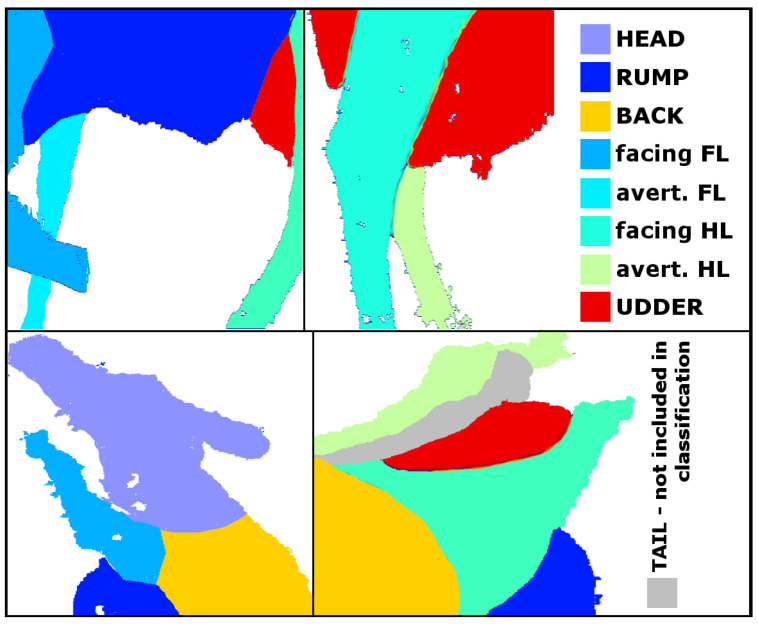
Example images from camera positions ‘S’ (top row) as well as ‘U’ and ‘N’ (bottom row). The image background was whitened and the result of labelling the images with the polygon drawer were displayed. Body parts were colored differently.

**Figure 3 animals-11-00050-f003:**
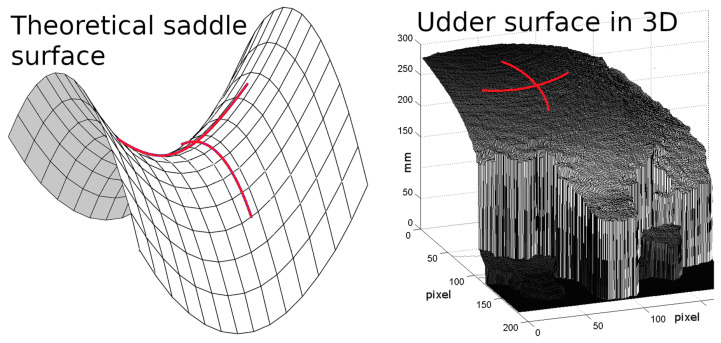
Exemplary visualisation of principal curvatures (red) in a point on a theoretical saddle surface (**left**) and the surface of an udder as measured by the Kinect (**right**). The mean curvature is the average of the principal curvatures, but can also be calculated from the gradient and the Hessian matrix in the respective surface point (Equation (Equation 1)).

**Figure 4 animals-11-00050-f004:**
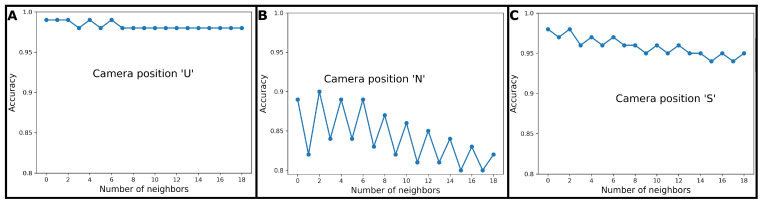
Accuracy depending on the number of neighbours in training a kNN classifier for camera positions ‘U’ (figure part **A**), ‘N’ (figure part **B**) and ‘S’ (figure part **C**). The hyperparameter k was tuned via grid search with five fold cross-validation.

**Figure 5 animals-11-00050-f005:**
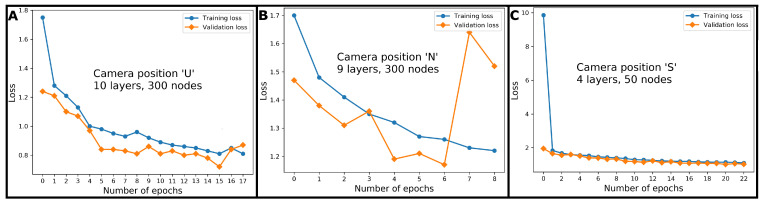
Training and validation loss depending on the number of trained epochs for camera positions ‘U’ (figure part **A**), ‘N’ (figure part **B**) and ‘S’ (figure part **C**).

**Table 1 animals-11-00050-t001:** Extract from the data set of camera position ’S’ before standardisation. The last eight columns provide the one-hot encoded body parts to which the pixels belong: head (He), rump (Ru), back (Ba), facing foreleg (fFL), averted foreleg (aFL), facing foreleg (fHL), averted hindleg (aHL), udder (Ud). Columns two to six hold the pixel properties row index, column index, depth value in mm, mean curvature (m_curv) and variance (var) both calculated from the surrounding nine-pixel-square.

No	Row	Col	Depth	m_curv	var	He	Ru	Ba	fFL	aFL	fHL	aHL	Ud
0	139	113	1050	0.66	5.4990	1	0	0	0	0	0	0	0
1	356	295	672	0.50	0.1945	0	1	0	0	0	0	0	0
2	366	46	931	−0.04	2.2500	1	0	0	0	0	0	0	0
3	352	60	746	0.36	0.6115	0	0	0	0	0	0	1	0
4	107	111	644	0.50	0.1109	0	1	0	0	0	0	0	0
5	110	267	1140	2.00	4.4437	1	0	0	0	0	0	0	0
6	326	160	801	1.00	0.7779	0	0	0	1	0	0	0	0
7	75	301	976	0.00	0.0000	0	1	0	0	0	0	0	0
8	283	388	1076	−0.05	2.2512	0	0	0	0	0	1	0	0
9	573	153	936	3.00	2.4996	0	0	1	0	0	0	0	0
10	460	244	1050	1.50	1.7503	1	0	0	0	0	0	0	0
11	544	297	1001	0.55	3.2509	0	1	0	0	0	0	0	0
12	235	93	685	0.00	0.1109	0	0	0	0	0	0	0	1
...			......						...	...			

**Table 2 animals-11-00050-t002:** Confusion matrix used to evaluate the classification of pixels.

	Pixel on body part = True	Pixel on body part = False
Pixel classified to be on body part = True	*TP*	*FP*
Pixel classified to be on body part = False	*FN*	*TN*

**Table 3 animals-11-00050-t003:** Evaluation metrics for the methods kNN classification and neural network classification (NN). Accuracy and Hamming loss (Ha.loss) were calculated for all classes together, while Precision, Recall and F1-score were calculated separately for the body parts head (He), rump (Ru), back (Ba), facing foreleg (fFL), averted foreleg (aFL), facing foreleg (fHL), averted hindleg (aHL) and udder (Ud) using the ‘one versus rest’ approach.

		knn								NN							
Accuracy	U	0.976								0.841							
	N	0.888								0.684							
	S	0.963								0.777							
Ha.loss	U	0.007								0.045							
	N	0.027								0.079							
	S	0.009								0.056							
		He	Ru	Ba	fFL	aFL	fHL	aHL	Ud	He	Ru	Ba	fFL	aFL	fHL	aHL	Ud
Precision	U	0.96	0.97	0.99	0.99	1.00	0.96	1.00	-	0.80	0.77	0.90	0.82	0.97	0.69	0.95	-
	N	0.93	0.87	0.77	0.89	0.92	0.90	0.95	0.84	0.82	0.55	0.97	0.58	0.73	0.68	0.63	0.53
	S	0.95	0.96	0.97	0.96	1.00	0.96	0.98	0.96	0.67	0.78	0.78	0.75	0.94	0.68	0.81	0.74
Recall	U	0.96	0.97	0.98	0.99	1.00	0.97	1.00	-	0.70	0.66	0.85	0.93	0.99	0.84	0.94	-
	N	0.95	0.84	0.98	0.87	0.96	0.88	0.93	0.83	0.71	0.60	1.00	0.70	0.70	0.57	0.71	0.48
	S	0.95	0.96	0.97	0.97	1.00	0.96	0.99	0.98	0.45	0.61	0.85	0.90	1.00	0.62	0.95	0.83
F1-score	U	0.96	0.97	0.99	0.99	1.00	0.97	1.00	-	0.75	0.71	0.87	0.87	0.98	0.76	0.94	-
	N	0.94	0.85	0.86	0.88	0.94	0.89	0.94	0.84	0.76	0.58	0.99	0.63	0.72	0.62	0.67	0.50
	S	0.95	0.96	0.97	0.97	1.00	0.96	0.98	0.97	0.54	0.69	0.81	0.82	0.97	0.65	0.88	0.78

## Data Availability

Raw data could not be made publicly available due to data privacy regulations with the participating farm.

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
