# Peer review of "Determination of Body Parts in Holstein Friesian Cows Comparing Neural Networks and k Nearest Neighbour Classification"

_animals, 2020, doi:10.3390/ani11010050_

Round 1

Reviewer 1 Report

The paper addresses the identification of cow body parts (Holstein-Freisian cows) from Kinect images using two ML techniques, namely KNN and neural networks. Indeed, there is an increasing interest among the ML community in applying algorithms to aid animal science. The paper is interesting and of practical importance. I am favourable towards further considering this paper for another review round provided some loose/weak points are addressed:

  • Kinect v1 is rather old hardware. The kinect v2 is even discontinued, and the kinect azure came late 2019. Have you considered using newer hardware, and what would be the impact on the approach?
  • The simple summary should motivate why it is important to recognize these cow parts in the first place. What is the purpose behind this approach?
  • Simple summary by the end: What applications do you have in mind? And which method would fit one or the other application and why?
  • Page 1, line 30: “were ranging” -> “ranged from”.
  • Page 2, line 60: How costly? Could you please provide some numbers and what would be the impact in practice of this inefficiency?
  • Page 2, line 80: You could also cite alongside [38] the following paper: “Estimating Body Condition Score in Dairy Cows from Depth Images Using Transfer Learning and Model Ensembling over Convolutional Neural Networks”. Agronomy - Special Issue on Deep Learning Techniques for Agronomy Applications. MDPI. 2019. This would be desirable in order to build the consistency of the MDPI journals.
  • Section 1, last paragraph: The structure of the paper should be explained here.
  • Page 3, line 102: The structure of the paper is confusing. Why placing the results section before explaining the approach, dataset and parameters employed? I also miss more context: why detecting body parts in the first place? What computational vision application are you pursuing?
  • Page 3, line 103: Same as above, At this point, camera positions and other details are just forward references.
  • Figure 1: What do these neighbors represent?
  • Page 3, line 108: How was the ground truth determined and represented? The pixel properties are not explained either. These results cannot be understood unless placed after the model and experiment conditions are explained.
  • Page 5, line 135: The paper should be self-contained. Please give further details on your decision.
  • Page 5, lines 135-136: I believe the paper would benefit from a Figure to explain the various interferences ocurring in your multi-kinect setting.
  • Page 5, line 142: which work?
  • Page 5, line 148: How so? Have you empirically compared both approaches?
  • Page 5, line 153 onwards: “The depth ...”. The five features used should be explained separately, perhaps using bullets or subsections. They represent the core of the ML approach!
  • Page 5, lines 155-165: The “structure” of the dataset is not clear. For every row in the training set, what is the information stored? I believe it is pixel metrics and the associated class (udder, rump, etc.). But I am not really sure. An example would help.
  • In general, Section 3 should be seriously revised. It spans three pages (out of 16 in the whole paper) and includes a lot of concepts, technical details and design decisions and it is easy to get lost while reading it.
  • Page 7, table 2: This should be a bullet list rather than a table. How many cameras did you use in total to support U, N and S? 2 (U) + 2 (N) + 2 (S)?
  • Section 4.2: What did extrinsic calibration involve? And what about intrinsics calibration? Instead of providing external differences, I suggest to explain briefly what was done. The same applies to the background substraction algorithm.
  • Page 8, line 293: “turned of” -> “turned off”?
  • Figure 3: Please place this figure closer to where it is referenced in the text.
  • Page 9, line 309: How are these curvatures determined, depending on the body part? Could you please visualize this somehow?
  • Conclusions: Would a different cow breed cause different results? Could you please comment on this issue?

Author Response

Dear Mrs., Mr,
Thank you for reviewing our article named ’Determination of body parts in Holstein Friesian cows comparing neural networks and k nearest neighbour classification’. I have reviewed your remarks and tried to make the paper stronger by taking into account your constructive comments on this paper. I am sure these comments improve the quality of the paper and I hope they meet your expectations. Below you can see my answers (italics) to your comments. The adjustments I made are highlighted in the resubmitted manuscript.
If there are any questions about them, I will be glad to answer them.

1. Comments and Suggestions for Authors The paper addresses the identification of cow body parts (Holstein-Freisian cows) from Kinect images using two ML techniques, namely KNN and neural networks. Indeed, there is an increasing interest among the ML community in applying algorithms to aid animal science. The paper is interesting and of practical importance. I am favourable towards further considering this paper for another review round provided some loose/weak points are addressed: Kinect v1 is rather old hardware. The kinect v2 is even discontinued, and the kinect azure came late 2019. Have you considered using newer hardware, and what would be the impact on the approach?
This reviewer is correct. The building of the prototype and the acquisition of data happened during the change from Kinect V1 to Kinect V2. We considered using the Kinect V2, but it was not possible to run several items of the Kinect V2 by one computer, as this was not supported by Microsoft. Thus, we stayed with the Kinect V1.

2. The simple summary should motivate why it is important to recognize these cow parts in the first place. What is the purpose behind this approach? Simple summary by the end: What applications do you have in mind? And which method would fit one or the other application and why?
Thank you for pointing this out. We added the purpose to the simple summary. The last sentence of the simple summary was shortend. The applications are no longer mentioned, as the simple summary is limited to 200 words.

3. Page 1, line 30: ?were ranging? -> ?ranged from?.
This was corrected.

4. Page 2, line 60: How costly? Could you please provide some numbers and what would be the impact in practice of this inefficiency?
The authors think, that this reviewer is here referring to the passage ’Although it served successfully as basis for the calculation of linear traits, this model based object detection was very costly in terms of computational time and memory.’ from the section Introduction. We added a sentence to this passage.

5. Page 2, line 80: You could also cite alongside [38] the following paper: ?Estimating Body Condition Score in Dairy Cows from Depth Images Using Transfer Learning and Model Ensembling over Convolutional Neural Networks?. Agronomy - Special Issue on Deep Learning Techniques for Agronomy  Applications. MDPI. 2019. This would be desirable in order to build the consistency of the MDPI journals.
The article suggested by this reviewer was cited.

6. Section 1, last paragraph: The structure of the paper should be explained here.
The end of the section ’Introduction’ was reworked and now gives more insight on the structure of the work and the objectives.

7. Page 3, line 102: The structure of the paper is confusing. Why placing the results section before explaining the approach, dataset and parameters employed? I also miss more context: why detecting body parts in the first place? What computational vision application are you pursuing?
The authors used the MDPI LATEX template in which materials and methods had been placed after the discussion. The section ’Materials and methods’ was now placed in front of the section ’Results’. Additional context was provided in the sections ’Introduction’.

8. Page 3, line 103: Same as above, At this point, camera positions and other details are just forward references. Figure 1: What do these neighbors represent?
See answer to comment 7.. The authors are confident, that the figure will be understandable since the Materials and methods section was re-placed.

9. Page 3, line 108: How was the ground truth determined and represented? The pixel properties are not explained either. These results cannot be understood unless placed after the model and experiment conditions are explained.
See answer to comment 7.. The authors are confident, that the results and the pixel properties will be understandable since the Materials and methods section was re-placed.

10. Page 5, line 135: The paper should be self-contained. Please give further details on your decision.
Thank you. Our decision is now explained in the text.

11. Page 5, lines 135-136: I believe the paper would benefit from a Figure to explain the various interferences ocurring in your multi-kinect setting.
The authors are confident, that Figure 1, A illustrates the positioning of the cameras and , thus, the lines of sights and the planes of projection of infrared pattern very clearly. Furthermore, the back-toback fixation of the devices in camera positions ’N’ and ’U’ including the angle bisector between the two fields of view is presented in Figure 1, B. Therefore, the readers could easily see in which sector interferences occur. An additional figure regarding the camera positions and lines of sight would in the authors opinion only burden the text. Figure 1 was referenced at the passage this reviewer is referring with in this comment.

12. Page 5, line 142: which work?
The respective reference was inserted.

13. Page 5, line 148: How so? Have you empirically compared both approaches?
The authors think, that this reviewer is referring to the sentence “With the object recognition presented here, a faster and more flexible approach is provided, as a larger variety of body parts are determinable for all three camera positions.”. Yes, both approaches have been developed and implemented by the authors.
It is now stated explicitly in the introduction, that the present study serves as a complement to the study in which the model based appraoch was introduced, and that it was the aim to provide a faster and more flexible solution with regard to the body part detection. The discussion was enhanced at this point to repeat this matter for the sake of clarity.

14. Page 5, line 153 onwards: ?The depth ...?. The five features used should be explained separately, perhaps using bullets or subsections. They represent the core of the ML approach!
The authors think, that this reviewer is here referring to the passage ’In this application five features are used. Row and column index localise the pixel in the depth map. These were considered, because it was likely for some body parts to occur in repeating pixel ranges, as exemplarily the udder in the recordings of sideview cameras mostly were depicted in the upper half of the depth map (Figure 4). The depth value was chosen as a feature, because differing body parts were recorded in different distances to the camera, i.e. back and rump or averted and facing legs. The mean curvature and the variance were both calculated for a square of surrounding pixels and describe characteristics of the recorded surface, as it was assumed that both the surface curvature as well as the texture differed between body parts.’ in the Discussion. Due to the former order of the section given by the template, this reviewer is right, that the mentioning of the features would have made more explanation necessary. Since the section ’Materials
and methods’ including a (reworked according to your comment 21.) definition of the pixel properties (=features) is now placed in front of the Discussion, the authors are confident, that this passage could be well understood.

15. Page 5, lines 155-165: The ?structure? of the dataset is not clear. For every
row in the training set, what is the information stored? I believe it is pixel metrics and the associated class (udder, rump, etc.). But I am not really sure. An example would help.
Due to the former order of the section given by the template, this reviewer is right, that the structure of the data set has not been explained in the Discussion section. Since the section ’Materials and methods’, actually including an extract from the data set as an example (Table 1) as well as detailed description, is now placed in front of the Discussion, the authors are confident, that the structure of the data set is now well known when reading the Discussion.

16. In general, Section 3 should be seriously revised. It spans three pages (out of
16 in the whole paper) and includes a lot of concepts, technical details and design decisions and it is easy to get lost while reading it.
The discussion was revised taking into account this reviewers valuable comments 10., 13. and 22.

17. Page 7, table 2: This should be a bullet list rather than a table. How many cameras did you use in total to support U, N and S? 2 (U) + 2 (N) + 2 (S)?
Table 2 was converted to a list of bullet points. Two cameras per position were used. This information was additionally placed below the bullet point list.

18. Section 4.2: What did extrinsic calibration involve? And what about intrinsics calibration? Instead of providing external differences, I suggest to explain briefly what was done. The same applies to the background substraction algorithm.
The authors did not want to burden the text with topic that had been published in detail in our earlier work. Nevertheless, in the respective paragraph, more information was given on the topics calibration and background subtraction.

19. Page 8, line 293: ?turned of? -> ?turned off??
This was corrected.

20. Figure 3: Please place this figure closer to where it is referenced in the text.
Whilst reworking the manuscript, Figure 3 was in a meaningful way placed as near as possible to its reference in the text.

21. Page 9, line 309: How are these curvatures determined, depending on the body part? Could you please visualize this somehow?
The mean curvature is defined mathematically as the average of the principal curvatures. The principal
curvatures are defined as the minimum and maximum Eigenvalues of the second fundamental form. The
mean curvature in a point of a surface is calculated - independently from the body part the point lies on
- by using the gradient and the Hessian matrix of the surface in this point. The formula was given and
a visualisation was added.

22. Conclusions: Would a different cow breed cause different results? Could you please comment on this issue?
The topic of a different breed was raised in the revised discussion and as a consequence in the conclusion as well.

Reviewer 2 Report

The author proposed a comparison between k-NN and ANN for detecting different body parts of dairy cows from MS-Kinect IR-sensor data using pixel-wise features.

My comments regarding this article are as follows:

  1. What I'm mostly missing is the research background, particularly in the introductory part. Why is it even important to recognize different body parts of dairy cows? I am not very convinced how this can be related to the welfare status solely based on sensor data? More appropriate literatures must to be added to justify the research background. Many applied machine learning & computer vision papers nowaday only introduced technicalities without having a proper justification why ML should even be used for solving certain problems.
  2. Technical presentations seem fine: I have experience working with both Kinect & machine learning, and to my knowlegde, the methodology sounds valid. As a side note however, they could have used ConvNet to increase the performance metrics (F1, ACC, etc.) by employing multi-modal sensor data. MS-Kinect has both IR-RGB camera, so ConvNet can technically be used to cross-correlate depth data and RGB data. The training can be then done in semi-supervised way (relying on available labels from IR to be used for RGB data - if this is not available yet).
  3. The hardware specification on which the algorithms were trained was missing. CPU / GPU? This should be specified because it is affecting the training performance.
  4. I'm not familiar with this journal style. Normally, scientific journal has following style: Introduction - Literature Reviews - Methods - Results - Discussions. Check w/ journal guideline.
  5. Spellings are fine, some minor checks needed, e.g. k-nearest neighbor can be abbreviated as kNN in the whole document to save a lot of spaces.
  6. As a conclusion, I can recommend revision before acceptance, but I would like to read more comprehensive research background justification.

Author Response

  Dear Mrs., Mr,
Thank you for reviewing our article named ’Determination of body parts in Holstein Friesian cows comparing neural networks and k nearest neighbour classification’. I have reviewed your remarks and tried to make the paper stronger by taking into account your constructive comments on this paper. I am sure these comments improve the quality of the paper and I hope they meet your expectations. Below you can see my answers (italics) to your comments. The adjustments I made are highlighted in the resubmitted manuscript. If there are any questions about them, I will be glad to answer them.

1. Comments and Suggestions for Authors The author proposed a comparison between k-NN and ANN for detecting different body parts of dairy cows from MS-Kinect IR-sensor data using pixel-wise features. My comments regarding this article are as follows: What I’m mostly missing is the research background, particularly in the introductory part. Why is it even important to recognize different body parts of dairy cows? I am not very convinced how this can be related to the welfare status solely based on sensor data? More appropriate literatures must to be added to justify the research background. Many applied
machine learning & computer vision papers nowaday only introduced technicalities without having a proper justification why ML should even be used for solving certain problems.
This reviewer is right, thank you for pointing this out. We improved the section ’Introduction’ and gave more background on the research question behind this article.

2. Technical presentations seem fine: I have experience working with both Kinect & machine learning, and to my knowlegde, the methodology sounds valid. As a side note however, they could have used ConvNet to increase the performance metrics (F1, ACC, etc.) by employing multi-modal sensor data. MS-Kinect has both IR-RGB camera, so ConvNet can technically be used to cross-correlate depth data and RGB data. The training can be then done in semi-supervised way (relying on available labels from IR to be used for RGB data - if this is not available yet). The hardware specification on which the algorithms were trained was missing. CPU / GPU? This should be specified because it is affecting the training performance.
The authors want to thank this reviewer for the suggestion. Furthermore, this reviewer is right, that the hardware specifications are missing. They are now provided in the section ’Materials and methods’.

3. I’m not familiar with this journal style. Normally, scientific journal has following style: Introduction -Literature Reviews - Methods - Results - Discussions. Check w/ journal guideline.
The authors used the MDPI LATEX template in which materials and methods had been placed after the discussion. The section ’Materials and methods’ was now placed in front of the section ’Results’.

4. Spellings are fine, some minor checks needed, e.g. k-nearest neighbor can be abbreviated as kNN in the whole document to save a lot of spaces. As a conclusion, I can recommend revision before acceptance, but I would like to read more comprehensive research background justification.
Thank you. We applied the abbreviation kNN throughout the script.

Reviewer 3 Report

General comments:

I find this paper very interesting as it uses relatively low-tech equipment while applying state of the art modelling and analyzing tools to generate a fascinating tool for body part determination in dairy cows. The technical and analytical framework is impressive and described in depth.

Many previous studies have tackled similar problems in many different ways with different types of 3D cameras, standard video cameras and digital cameras and of course the Kinect camera system. However, those studies have had a specific goal in mind, such as the goal of accurate and precise body condition scoring.

This manuscript is currently lacking a specific objective or goal more than what is generally mentioned in the title. I would like to see a description of a particular practical use of this methodology in the manuscript. What should this technology be used for? Is it to investigate the possibility of cow body conformation detection? Is it the investigation and development of the methodology comparison (software and analyzing tools) itself? Is it both?

What is the end goal here? Could it be used in the long run to automatically detect a teat deformity when cows passes a stationary setup? Is it a novel body condition scoring? What data is being collected and how useful is it for the producer? ? Is it a commercially viable product? If so, is it practical to install on farm? The practical use of the methodology should be determined and be discussed in the discussion.

In addition, it would be nice to have an objective in the abstract, or at least in the end of the introduction and preferably be noted as achieved or not in the conclusion.

My other general comment is that I find it odd to have the material and methods coming after the discussion. I would move it up to before the results section and change header and sub-header numbers accordingly.

More specific comments are found below:

Abstract

Ln 33-34: Consider rephrasing last sentence slightly to avoid using the reference to  “This”.  Perhaps along the lines of: “The cost vs accuracy ratio for each methodology needs to be taking into account in the decision which method should be implemented in the application”

Introduction

Ln 38-39: Misspelled “livestock”.

Ln 39-40: Rephrase start of sentence from “It is the goal…” to “The goal of precision livestock farming is to inform farmers…”

Ln 43-44: Consider adding “Common…” to start of sentence and change “were” to “are” as in “are diffused light conditions….”

Ln 73-76: Consider rewording to avoid starting sentence with citation or write out study author and add citation in the end of sentence.

Ln 78-79: “…, neural networks seem a suited tool..), add “seems to be a suited tool ([36]).”

Ln 80: This sentence seems currently to be free-standing and needs to be worked into the flow of the paragraph.  Perhaps add “For instance, “ in the beginning as in: “For instance, milk yield have been successfully predicted by the use of neural networks [37]. Furthermore…”

Ln 84-85: Here and elsewhere, same comment as Ln 73-76.

Material and methods

Ln 259-260: There is currently no figure 3 attached to the manuscript. I also don’t understand the wording “framework” but maybe its clearer when the figure is present. Is it a structure like a passageway, alleyway in a barn or like a cattle chute?

Ln 261-263: Consider rewording that the two Kinect cameras were “…installed at a height of 0.6m above the ground…

Ln 273-274: What were the main limitations (if any) on the low pixel resolution of the Kinect system? Would the system have benefitted by having a higher resolution camera? Would accuracy of the system have improved by the ability to more exactly filter out background noise or would it have caused more issues of high-resolution blending of the cows body with the background? In addition, would a higher resolution enabled better distinction of the different body regions or would it have to be weighed against a larger number of data points to be run through the algorithm, thus taking more time and resources and extra cost involved?

Ln 289-290: Did you encounter any accuracy differences between younger (lactation 1) and older animals?

Ln 295-299: Out of curiosity, did you need to obtain an pre-approved animal study protocol for this study through the university under German Law or did you just have to adhere to the German Animal Welfare Act regulations?

Figure 4: Is it possible to rotate bottom panes to have the legs be pointing downwards to be consistent with the two top panes? I think that would make the figure more clear.

Ln 349-350: Could you explain the plain stack of dense layers in more detail. For a layman reader, this is hard to understand.

I feel that the Kruskal-Wallis test gets a bit lost in the end of the material and methods. Could more information about the test be provided (such as why it was chosen, non-parametric data or that the data was not assumed to be normally distributed etc) and put in its own sub-header?

Results

Figure 1 states ‘camera type’ but we are discussing camera positions, correct? Please change to clarify to the reader to avoid confusion,

Why was camera position S using 50 nodes per hidden layer? It is currently not explained.

Figure 2. An explanation of ‘epoch’ as the number of of complete passes of the entire training dataset would be good to add, preferably in the material and methods section to give the reader a better understanding of the methodology and terminology.

Discussion

Ln 134-135: Could a brief reason be given here so the reader does not have to go to another paper to find the rationale?

Ln 136-141: This reads more like a material and methods rather than discussion, I would consider moving this to the material and methods section. The camera placement due to inference and the 2 degree overlap is not mentioned there.

Ln 142-177: A large part of this section is describing methodology rationale. I would look over this a bit closer and see what would be better used as a section of rationale and methodology framework in the material and methods section and what discusses the results of the study. I would ask the authors to also see if any additional comparison of your results with other Kinect based studies can be made?

Ln 209-210. This is new information not currently found in the material and methods section.

Conclusions:

Currently, only lines 400-403 reads as a conclusion. The rest of the conclusion is a short summary of the paper.

What is the take home message of the study? What were the key findings? What can it be used for? Why is this important? What is improved from previous studies with similar technology?

Author Response

Dear Mrs., Mr,
Thank you for reviewing our article named ’Determination of body parts in Holstein Friesian cows comparing neural networks and k nearest neighbour classification’. I have reviewed your remarks and tried to make the paper stronger by taking into account your constructive comments on this paper. I am sure these comments improve the quality of the paper and I hope they meet your expectations. Below you can see my answers (italics) to your comments. The adjustments I made are highlighted in the resubmitted manuscript.
If there are any questions about them, I will be glad to answer them.

1. Comments and Suggestions for Authors General comments: I find this paper
very interesting as it uses relatively low-tech equipment while applying state of the art modelling and analyzing tools to generate a fascinating tool for body part determination in dairy cows. The technical and analytical framework is impressive and described in depth. Many previous studies have tackled similar problems in many different ways with different types of 3D cameras, standard video cameras and digital cameras and of course the Kinect camera system. However, those studies have had a specific goal in mind, such as the goal of accurate and precise body condition scoring.
This manuscript is currently lacking a specific objective or goal more than what is generally mentioned in the title. I would like to see a description of a particular practical use of this methodology in the manuscript. What should this technology be used for? Is it to investigate the possibility of cow body conformation detection? Is it the investigation and development of the methodology comparison
(software and analyzing tools) itself? Is it both? What is the end goal here? Could it be used in the long run to automatically detect a teat deformity when cows passes a stationary setup? Is it a novel body condition scoring? What data is being collected and how useful is it for the producer? ? Is it a commercially viable product? If so, is it practical to install on farm? The practical use of the methodology should be determined and be discussed in the discussion.
In addition, it would be nice to have an objective in the abstract, or at least in the end of the introduction and preferably be noted as achieved or not in the conclusion.
The end of the section ’Introduction’ was reworked and now gives more insight on the structure of the work and the objectives.

2. My other general comment is that I find it odd to have the material and methods coming after the discussion. I would move it up to before the results section and change header and sub-header numbers accordingly.
The authors used the MDPI LATEX template in which materials and methods had been placed after the discussion. The section ’Materials and methods’ was now placed in front of the section ’Results’.

3. More specific comments are found below: Abstract Ln 33-34: Consider rephrasing last sentence slightly to avoid using the reference to ?This?. Perhaps along the lines of: ?The cost vs accuracy ratio for each methodology needs to be taking into account in the decision which method should be implemented in
the application?
Thank you for this valuable suggestion. The passage was rephrased.

4. Introduction Ln 38-39: Misspelled ?livestock?.
Thank you, the typo was corrected.

5. Ln 39-40: Rephrase start of sentence from ?It is the goal?? to ?The goal of precision livestock farming is to inform farmers??
Thank you for this valuable suggestion. The passage was rephrased.

6. Ln 43-44: Consider adding ?Common?? to start of sentence and change ?were? to ?are? as in ?are diffused light conditions?.?
Thank you for this valuable suggestion. The passage was rephrased.

7. Ln 73-76: Consider rewording to avoid starting sentence with citation or write out study author and add citation in the end of sentence.
The sentence was rephrased.

8. Ln 78-79: ??, neural networks seem a suited tool..), add ?seems to be a suited tool ([36]).?
This was corrected.

9. Ln 80: This sentence seems currently to be free-standing and needs to be worked into the flow of the paragraph. Perhaps add ?For instance, ? in the beginning as in: ?For instance, milk yield have been successfully predicted by the use of neural networks [37]. Furthermore??
Thank you for this valuable suggestion. The passage was rephrased.

10. Ln 84-85: Here and elsewhere, same comment as Ln 73-76.
The sentence was rephrased.

11. Material and methods Ln 259-260: There is currently no figure 3 attached to the manuscript. I also don?t understand the wording ?framework? but maybe its clearer when the figure is present. Is it a structure like a passageway, alleyway in a barn or like a cattle chute?
The authors are deeply sorry. It is unclear to us, why no figure 3 was attached in the manuscript this reviewer got to read. Figure 3 has been included in the pdf build by LaTex. It showed the recording unit and the combined field of view of the cameras.

12. Ln 261-263: Consider rewording that the two Kinect cameras were ??installed at a height of 0.6m above the ground?
Thank you for your suggestion, the sentence was rephrased.

13. Ln 273-274: What were the main limitations (if any) on the low pixel resolution of the Kinect system? Would the system have benefitted by having a higher resolution camera? Would accuracy of the system have improved by the ability to more exactly filter out background noise or would it have caused more
issues of high-resolution blending of the cows body with the background? In addition, would a higher resolution enabled better distinction of the different body regions or would it have to be weighed against a larger number of data points to be run through the algorithm, thus taking more time and resources
and extra cost involved?
The authors want to thank this reviewer for these important questions and included these topics in the Discussion.

14. Ln 289-290: Did you encounter any accuracy differences between younger (lactation 1) and older animals?
Thank you for this interesting consideration. However, the number of animals (18) would have been not large enough and the groups would have been highly unbalanced, so that no basis for a stressable statistical comparison was present. Furthermore, separate training of the algorithms would have necessarily been
taken place on separate data sets within the respective lactation number. Thereby the number of data point used for training would have been smaller than training without grouping, thus accuracy would likely have been smaller for all lactation number due to smaller training data sets. In addition, it is questionable, whether the data sets within lactation number would have been comparably large. Although an interesting question, – and of course a possible effect on the accuracy – this analysis has not been the goal of the present manuscript. A special design starting at data collection would have been necessary to ensure comparability in the number of animals per lactation number as well as the sizes of the training
data sets.

15. Ln 295-299: Out of curiosity, did you need to obtain an pre-approved animal study protocol for this study through the university under German Law or did you just have to adhere to the German Animal Welfare Act regulations?
We only needed to adhere to the regulations of the ’German Animal Welfare Act’ and the ’German Order for the Protection of Animals used for Experimental Purposes and other Scientific Purposes’.

16. Figure 4: Is it possible to rotate bottom panes to have the legs be pointing downwards to be consistent with the two top panes? I think that would make the figure more clear.
Thank you for this suggestions. However, the subfigures in the bottom row of the respective figure (former Figure 4) show exactly, how the top view cameras see the passing cows. Therefore, the authors consider it valuable to keep the images in it original orientation for a better understanding of both the cameras angle of view towards the animals and the body parts to be determined as well as the recording unit and the character of the data collected.

17. Ln 349-350: Could you explain the plain stack of dense layers in more detail. For a layman reader, this is hard to understand.
Explanation was added. Thank you.

18. I feel that the Kruskal-Wallis test gets a bit lost in the end of the material and methods. Could more information about the test be provided (such as why it was chosen, non-parametric data or that the data was not assumed to be normally distributed etc) and put in its own sub-header?
Information on the Kruskal Wallis test and its application were added and placed within its own paragraph.

19. Results Figure 1 states ?camera type? but we are discussing camera positions, correct? Please change to clarify to the reader to avoid confusion,
Thank you for pointing this out. The figure was renewed.

20. Why was camera position S using 50 nodes per hidden layer? It is currently not explained.
As stated in former section 4.5.2. Neural networks (now section 2.5.2): “The hyperparameters ’numbers of hidden layers’ and ’number of neurons’ in the hidden layers were optimized within the following ranges
->number of hidden layers0 ∈ {0, ..., 16}
->number of neurons0 ∈ {10, 50, 100, 150, 200, 250, 300}.
Neural networks were trained and evaluated FOR ALL PAIRS OF hyperparameter values. In addition, early stopping after five epochs without improvement regarding the loss on the validation data as well as a scheduler to linearly decrease the model?s learning rate with increasing number of trained
epochs were implemented for every neural network model. The neural networks were chosen with regard to the smallest observed validation loss.” This means, that the 50 nodes per hidden layer associated with
camera position ’S’ have not been the authors’ choice, but the networks for all camera positions were

optimized on the above mentioned parameter grid. Thus, it is stated in the section ’Results’ “Regarding camera position ’S’ the validation loss reached its minimum for a neural network with four hidden layers and fifty nodes per hidden layer.” It is beyond the authors’ knowledge why the optimal network regarding the camera position ’S’ has fifty compared to 300 nodes per hidden layer. Neural networks are a highly flexible tool and with a large number of possibilities to be adapted to a specific application, e.g. via the main hyperparameters the number of hidden layers (depth) and the number of nodes (height). The choice of capacity is discussed in the section ’Discussion’, however, we additionally stressed the noteworthy difference between camera positions when it comes to the outcome of the optimizing of hyperparameters.

21. Figure 2. An explanation of ?epoch? as the number of of complete passes of the entire training dataset would be good to add, preferably in the material and methods section to give the reader a better understanding of the methodology and terminology.
Thank you for pointing this out. Explanation was added in the section ’Materials and methods’.

22. Discussion Ln 134-135: Could a brief reason be given here so the reader does not have to go to another paper to find the rationale?
The authors consider, that this reviewer is referring to the passage “Additional models would have had to be used with every additional camera position or body part. With the object recognition presented here, a faster and more flexible approach is provided, as a larger variety of body parts are determinable for all three camera positions.”, which starts at the end of line 133 and ends in line 136 in the pdf version of manuscript as revised. We are not sure, for what this reviewer is suggesting to give a reason. The complete passage starts with “The recording unit has in the aforementioned work been successfully applied to measure the conformation traits ’udder depth’ and ’rear leg angle’. The recognition of body parts, however, has been dealt with via a computationally very expensive object recognition pipeline based on models of the respective body parts, i.e. udder and hindleg. The model based object recognition has only been applied to the sideview cameras and only to udder and hindleg.” Reading the complete passage, the reasons for “Additional models would have had to be used...” and “[...] larger variety of body parts are determinable for all three camera positions.” were anticipated as well as the explanation for “[...] a faster and more flexible approach [...]”. The authors inserted the respective reference in the first sentence of the complete passage to make it more clear to which work we are referring.

23. Ln 136-141: This reads more like a material and methods rather than discussion, I would consider moving this to the material and methods section. The camera placement due to inference and the 2 degree overlap is not mentioned there.
Thank you for your valuable comment. However, the authors consider the section ’Material and methods’ to include solely the description of methods (and material) used, whilst the passage this reviewer is referring to serves the purpose of discussing our choice of features. The authors are confident, that this is not misplaced. Furthermore, the features are described in detail in the section  ’Materials and methods’ (which has been reworked to deliver a more detailed representation of the features).

24. Ln 142-177: A large part of this section is describing methodology rationale. I would look over this a bit closer and see what would be better used as a section of rationale and methodology framework in the material and methods section and what discusses the results of the study. I would ask the authors to also see if any additional comparison of your results with other Kinect based studies can be made?
This reviewer is correct, that this passage refers to methodology, however, the methodology is not introduced here, but its choice and application is discussed, after the methodology was properly introduced in the section ’Materials and methods.’ For example: It was explained how the SMOTE balancing technique
is generating new features, which implementation was used and to what part of the data it was applied in ’Material and methods’, whilst in the passage this reviewer is referring to the authors elaborated on potential problems and specifications regarding SMOTE which clearly belongs to the ’Discussion’. The
same applies to the discussion on the choice of machine learning algorithms. However, the respective passage was shortened to avoid double information. In the section ’Discussion’ the presented application was additionally compared to some other Kinect based studies.

25. Ln 209-210. This is new information not currently found in the material and methods section.
This reviewer is right. However, the section ’Materials and methods’ should only contain the information needed to generate and technically understand the results. When it comes to the used features, all the information necessary with regard to this definition of ’Materials and methods’ are provided there. The sentence this reviewer is referring to, however, serves as a probable explanation of the results and is not misplaced in the discussion. With regard to this reviewers comments 23.-25., the authors want to draw this reviewers attention to the authors guidelines of MDPI animals to demonstrate that our answers to the respective comments are reasonable: “Materials and Methods: They should be described with sufficient detail to allow others to replicate and build on published results. [...] . Give the name and version of any software used and make clear whether computer code used is available. [...]”

26. Conclusions: Currently, only lines 400-403 reads as a conclusion. The rest of the conclusion is a short summary of the paper. What is the take home message of the study? What were the key findings? What can it be used for? Why is this important? What is improved from previous studies with similar technology?
It says in the authors guidelines of MDPI animals: “Conclusions: This section is mandatory, and should provide readers with a brief summary of the main conclusions.”. Thus, a summary of the paper needs to be included into conclusions, as it could well be, that someone only reads the conclusion to save time in finding out whether it could serve his/her purpose to look into the article more deeply. This reviewer is, however, right, that additional questions could have been answered in the conclusion. It was reworked.

Reviewer 4 Report

The authors have used KNN and neural network models to body parts of cows. The choice of the models seems logical and overall, the paper described the methods well. Below are some suggestions for improvement and questions.

Simple summary

“Both algorithms used for this feature based approach are computationally faster and more flexible compared to a model based object detection”.

Q: Is a neural network just another model based object detection? Perhaps it is best to be specific on what it is faster and more flexible against. It will be good to compare the performance to an existing model used in the industry, if it exists.

Abstract

“In this study, two machine learning approaches were utilized for the recognition of  body parts of dairy cows from 3D point clouds.”

Q: If it is not already defined, please ensure readers understand what ‘3D point clouds’ are.

Introduction

“In lameness detection in cattle 2D cameras ([5]; [6]), 3D cameras ([7]; [8]; [9]) and combinations between cameras and other sensor data ([10]) came to use. [11] used …”

Q: Not sure if the sentence is meant to be one or two sentences. It does not read well with the current referencing style.

“Cameras have also been applied in animal identification ([17]), the monitoring herd activities, ([18]), animal behaviour ([19]; [20]) and the animals’ use of barn space ([21]).”

Q: Suggested improvement: “… animal identification (ref), monitoring of herd activities, (ref), animal behaviour …”

“([22] presented a scanning passage for dairy cows in which six Kinect V1 cameras were mounted.”

Q: Again, this referencing style makes it hard to read. Perhaps try to rewrite it in a different way to suit the referencing style. Ditto to the rest of the paper where you have used this style.

“The used Microsoft Kinect camera …”

Q: Suggested improvement: “The use of Microsoft …”

“… this model based object detection was very costly in terms of computational  time and memory.”

Q: How costly exactly in minutes and Gb of memory? Given the advances in computation and memory, is this still an issue?

Results

In Figure 1, the camera type N has higher accuracy whenever the number of neighbours is even rather than odd. This seems rather strange. Why?

The current Results section could be expanded to elaborate more results. For example, what is the compute time and memory differences between KNN, neural network and the existing method prior to this study?

Discussion

“Non-linear relations and a high number of parameters make a lot of problems associated with …”
Q: Please consider breaking this paragraph into two paragraphs.

“Thus this feature space might have been prone to a classification via the euclidean distance.”

Q: Suggested improvement: “Therefore, the features could be better modelled as a classification problem that uses the Euclidean distance.”

Methods

In general, for the method section, I’ll advise to be specific on the number of response and predictors. Additionally, the number of observations should be specified to give readers a sense of the size of the dataset.

“The hyperparameters ’numbers of hidden layers’ and ’number of neurons’ in the hidden layers were optimized within the following ranges: …”

Q: How is this optimised? Based on higher accuracy (defined as number of correct predictions/ total predictions)? The range of hidden layers and number of neurons, how were these chosen? As it is unclear how many predictors in this case, it is best to indicate what rules that you have chosen to optimise this. Did you use any back propagation algorithm? If yes, which one?

Author Response

Dear Mrs., Mr,
Thank you for reviewing our article named ’Determination of body parts in Holstein Friesian cows comparing neural networks and k nearest neighbour classification’. I have reviewed your remarks and tried to make the paper stronger by taking into account your constructive comments on this paper. I am sure these comments improve the quality of the paper and I hope they meet your expectations. Below you can see my answers (italics) to your comments. The adjustments I made are highlighted in the resubmitted manuscript.
If there are any questions about them, I will be glad to answer them.

1. Comments and Suggestions for Authors The authors have used KNN and neural
network models to body parts of cows. The choice of the models seems logical and overall, the paper described the methods well. Below are some suggestions for improvement and questions. Simple summary “Both algorithms used for this feature based approach are computationally faster and more flexible compared to a model based object detection”.
Q: Is a neural network just another model based object detection? Perhaps it is best to be specific on what it is faster and more flexible against. It will be good to compare the performance to an existing model used in the industry, if it exists.
As the simple summary should only contain 200 words, it is not feasible to compare models used in the industry with the models presented in this article. However, we tried to be more specific on the model based approach.

2. Abstract “In this study, two machine learning approaches were utilized for the
recognition of body parts of dairy cows from 3D point clouds.”
Q: If it is not already defined, please ensure readers understand what ?3D point clouds? are.
An explanation was added to the text.

3. Introduction “In lameness detection in cattle 2D cameras ([5]; [6]), 3D cameras ([7]; [8]; [9]) and combinations between cameras and other sensor data ([10]) came to use. [11] used ...”
Q: Not sure if the sentence is meant to be one or two sentences. It does not read well with the current referencing style.
Thank you for pointing out, that this passage was hard to follow. These are two sentences. The second sentence was now rephrased in a way, that it does not start with a reference. The authors are confident, that readability was improved by this.

4. “Cameras have also been applied in animal identification ([17]), the monitoring herd activities, ([18]), animal behaviour ([19]; [20]) and the animals? use of barn space ([21]).”
Q: Suggested improvement: “... animal identification (ref), monitoring of herd activities, (ref), animal behaviour ...
The authors are truly sorry, but we cannot find the structural difference between the sentence from the manuscript and the suggested improvement. In case, this reviewer is referring to the referencing style: The authors used the MDPI template. The references are set is by and according to this template.

5. “([22] presented a scanning passage for dairy cows in which six Kinect V1 cameras were mounted.”
Q: Again, this referencing style makes it hard to read. Perhaps try to rewrite it in a different way to suit the referencing style. Ditto to the rest of the paper where you have used this style.
The sentence was rephrased in a way, that it does not start with a reference any more.

6. “The used Microsoft Kinect camera ...”
Q: Suggested improvement: “The use of Microsoft ...”
Thank you for the suggestion. However, the sentence aims at describing the camera itself (measurement principle) and not its use.

7. “... this model based object detection was very costly in terms of computational time and memory.”
Q: How costly exactly in minutes and Gb of memory? Given the advances in computation and memory, is this still an issue?
Thank you for pointing this out. The authors added information on the computational time and the practical applicability to the text.

8. Results In Figure 1, the camera type N has higher accuracy whenever the number of neighbours is even rather than odd. This seems rather strange. Why?
This reviewer is right, that the behaviour differs between camera positions and is especially noteworthy for camera position ’N’. A range of values for the hyperparameter k was tested, and the structure of the data set and of the knn models were the same for all camera positions. Reasons due to the method or the structure of the data could, thus, be excluded. It has to be stated, that with  amera position ’S’ the accuracy is also going up and down with the change from odd to even values for k, although the amplitude is smaller. In general tie situations can happen when an even value is chosen for k, which might be the reason and was now included in the discussion. It is, however, unknown to the authors why this phenomenon occurred so strongly with camera position ’N’.

9. The current Results section could be expanded to elaborate more results. For example, what is the compute time and memory differences between KNN, neural network and the existing method prior to this study?
Thank you for this suggestion. The problem is that the run time of the pre-existing method strongly varies with the image computed, as several positions for the models to fit are tested. The information that it could take several minutes to process a single image has been included in the text, however, a descriptive analysis on computation time regarding a method which is not the main topic here, seems to burden the text. Furthermore, computation times are hardly comparable between the actual and the pre-existing methods, as the model based approach only delivers results for two body parts and the methods presented
here segment the complete cow (tail excluded). As a matter of fact, the hyperparameter specification as well as the training of the final methods have not been run on the same system and thus the computation times are not comparable either.

10. Discussion “Non-linear relations and a high number of parameters make a lot of problems associated with ...”
Q: Please consider breaking this paragraph into two paragraphs.
This paragraph was split.

11. “Thus this feature space might have been prone to a classification via the euclidean distance.”
Q: Suggested improvement: “Therefore, the features could be better modelled as a classification problem that uses the Euclidean distance.”
Thank you for your valuable suggestion. The sentence was rephrased.

12. Methods In general, for the method section, I?ll advise to be specific on the number of response and predictors. Additionally, the number of observations should be specified to give readers a sense of the size of the dataset.
The authors made modifications to paragraph 2.4 (former 4.4) to give the information on the number of response and predictors more clearly. In addition, we stated this again at the beginning of paragraph 2.5 (former 4.5). Nevertheless, it says in former lines 337-339: “The input layer of all models held five neurons, one for each pixel property (section 4.4, Table 3). The output layer of all models held seven (camera position ’U’), respectively, eight nodes (camera positions ’N’ and ’S’) according to the number of body parts to distinguish between.” This information was kept there.. ’Additionally, the number of observations should be specified to give readers a sense of the size of the dataset.’: The authors are sorry, however, this information is stated clearly in the manuscript: It says
in former lines 298-300 (now lines 175-177) “One list per camera position was generated resulting in 915,587 rows for camera position ’U’, 1,295,302 rows for camera position ’N’, and 1,352,017 rows for camera position ’S’.”, hereby specifying the number of observations for all camera types.

13. “The hyperparameters ?numbers of hidden layers? and ?number of neurons? in the hidden layers were optimized within the following ranges: ...”
Q: How is this optimised? Based on higher accuracy (defined as number of correct predictions/ total predictions)? The range of hidden layers and number of neurons, how were these chosen? As it is unclear how many predictors in this case, it is best to indicate what rules that you have chosen to optimise this. Did you use any back propagation algorithm? If yes, which one?
The neural networks were chosen with regard to the smallest observed validation loss, as stated in former
lines 352/353. This statement was placed nearer to the passage this reviewer was referring to to make
the optimisation more clear. The authors unfortunately have to disagree with this reviewer, as it had
already been explained prior to the definition of the hyperparameter range, that all networks had five
nodes in their input layer (predictors) and seven/eight nodes in their output layer (responses). (Also compare our answer to your comment 12.) As the needed capacity for the problem at hand was unknown, and the ranges within the hyperparameters were optimised had to be chosen by the user, the ranges for
nuber of hidden lyers and number of neurons were chosen to provide a wide variety architectures of fully connected neural network. This additional explanation was added to the text. The models were trained using backpropagation as it is the default in Keras Sequential() Models .fit function. This was added to the manuscript. The specifications were given by the loss function and the optimizer, these information were already present in the text.

Round 2

Reviewer 1 Report

I am happy that the authors have seriously addressed all my comments. Some further suggestions to polish the paper are:
- With respect to your reply "The building of the prototype and the acquisition of data happened during the change from Kinect V1 to Kinect V2. We considered using the Kinect V2, but it was not possible to run several items of the Kinect V2 by one computer, as this was not supported by Microsoft. Thus, we stayed with the Kinect V1.". I suggest add this to the paper somewhere, so readers are not caught by surprise when reading the paper considering that we have the Kinect v3 around us already.
- Reference 14 is incomplete. I believe it is "Estimating Body Condition Score in Dairy Cows from Depth Images Using Transfer Learning and Model Ensembling over Convolutional Neural Networks. Agronomy - Special Issue on Deep Learning Techniques for Agronomy Applications. MDPI. 2019."
- With respect to my comment "Section 1, last paragraph: The structure of the paper should be explained here.". I appreciate the text added, but could you please also point to the Sections including this information?
- I also thank authors for taking the time to reorder sections. I believe this greatly improves the flow of the text. I suggest authors to carefully check for erroneous backward/forward to pieces of information (if any) upon preparing the final version.

Author Response

Dear Mrs., Mr,
Thank you for the second round review of our article named ’Determination of body parts in Holstein Friesian cows comparing neural networks and k nearest neighbour classification’. I have reviewed your remarks and polished the paper according to your suggestions. Below you can see my answers (italics) to your
comments. The adjustments I made are highlighted in the resubmitted manuscript. If there are any questions about them, I will be glad to answer them.

1. I am happy that the authors have seriously addressed all my comments. Some further suggestions to polish the paper are: - With respect to your reply “The building of the prototype and the acquisition of data happened during the change from Kinect V1 to Kinect V2. We considered using the Kinect V2, but it was not possible to run several items of the Kinect V2 by one computer, as this was not supported by Microsoft. Thus, we stayed with the Kinect V1.”. I suggest add this to the paper somewhere, so readers are not caught by surprise when reading the paper considering that we have the Kinect v3 around us already.

Thank you for pointing this out, we added this information to the introduction to avoid irritation on
this matter.

2. - Reference 14 is incomplete. I believe it is "Estimating Body Condition Score in Dairy Cows from Depth Images Using Transfer Learning and Model Ensembling over Convolutional Neural Networks. Agronomy - Special Issue on Deep Learning Techniques for Agronomy Applications. MDPI. 2019."

We completed the reference.

3. - With respect to my comment "Section 1, last paragraph: The structure of the paper should be explained here.". I appreciate the text added, but could you  please also point to the Sections including this information?

Tank you for your suggestion. The references were included.

4. - I also thank authors for taking the time to reorder sections. I believe this greatly improves the flow of the text. I suggest authors to carefully check for erroneous backward/forward to pieces of information (if any) upon preparing the final version.

This reviewer is right, that readability was improved. Authors took care of backward and forward references. Thank you again for reviewing.

Reviewer 2 Report

The authors have modified and improved the manuscript as suggested, and these includes more sufficient research background as well as the specification of the hardwares used to train the model.

To my knowledge, the manuscript can now be accepted.

Author Response

The authors would like to thank this reviewer for the second round review of our paper "Determination of body parts in Holstein Friesian cows comparing neural networks and k nearest neighbour classification". We are happy to hear, that this reviewers expectations have been met by our latest improvements.